# Changes in Epigenetic Patterns Related to DNA Replication in *Vicia faba* Root Meristem Cells under Cadmium-Induced Stress Conditions

**DOI:** 10.3390/cells10123409

**Published:** 2021-12-03

**Authors:** Aneta Żabka, Natalia Gocek, Konrad Winnicki, Paweł Szczeblewski, Tomasz Laskowski, Justyna Teresa Polit

**Affiliations:** 1Department of Cytophysiology, Faculty of Biology and Environmental Protection, University of Lodz, 90-236 Lodz, Poland; natalia.gocek@edu.uni.lodz.pl (N.G.); konrad.winnicki@biol.uni.lodz.pl (K.W.); justyna.polit@biol.uni.lodz.pl (J.T.P.); 2Department of Pharmaceutical Technology and Biochemistry, Faculty of Chemistry, Gdansk University of Technology, 80-233 Gdansk, Poland; pawel.szczeblewski@pg.edu.pl (P.S.); tomasz.laskowski@pg.edu.pl (T.L.)

**Keywords:** cadmium, DNA methylation, DNA replication, epigenetic modifications, H3K56Ac, H3K79Me2, H3T45Ph, replication stress, transcription, *Vicia faba*

## Abstract

Experiments on *Vicia faba* root meristem cells exposed to 150 µM cadmium chloride (CdCl_2_) were undertaken to analyse epigenetic changes, mainly with respect to DNA replication stress. Histone modifications examined by means of immunofluorescence labeling included: (1) acetylation of histone H3 on lysine 56 (H3K56Ac), involved in transcription, S phase, and response to DNA damage during DNA biosynthesis; (2) dimethylation of histone H3 on lysine 79 (H3K79Me2), correlated with the replication initiation; (3) phosphorylation of histone H3 on threonine 45 (H3T45Ph), engaged in DNA synthesis and apoptosis. Moreover, immunostaining using specific antibodies against 5-MetC-modified DNA was used to determine the level of DNA methylation. A significant decrease in the level of H3K79Me2, noted in all phases of the CdCl_2_-treated interphase cell nuclei, was found to correspond with: (1) an increase in the mean number of intranuclear foci of H3K56Ac histones (observed mainly in S-phase), (2) a plethora of nuclear and nucleolar labeling patterns (combined with a general decrease in H3T45Ph), and (3) a decrease in DNA methylation. All these changes correlate well with a general viewpoint that DNA modifications and post-translational histone modifications play an important role in gene expression and plant development under cadmium-induced stress conditions.

## 1. Introduction

Due to a sedentary lifestyle, plants are constantly exposed to a variety of environmental stresses which largely hinder their proper development. In addition to a number of biotic factors, e.g., pathogens, pests, or plant competitiveness, many abiotic stressors, such as too high or low temperature, severe shortage of water, lack or excess of minerals, salinity, intense solar radiation, and heavy metals, represent significant limiting factors for plants’ growth and productivity [1,2,3]. Therefore, in order to prevent stressful situations, plants have developed complicated and complex mechanisms for stress detection and defense reactions [4]. Accordingly, they activate various signal transduction pathways to allow the expression of stress-responsive genes, which ultimately induce changes at the morphological, physiological, and biochemical levels that adapt plants to adverse environmental conditions [5,6].

The mechanism of genome-wide DNA duplication precisely carries out its functions despite numerous obstacles of intracellular and extracellular origin, many of which can lead to “replication stress” (RS). It is well known that there are many sources of RS and, thus, its definition is constantly evolving and difficult to formulate. Currently, we define RS as slowing down or stopping the movement of replication forks and/or DNA synthesis. Since DNA synthesis, as well as the cellular response to RS, occur in the context of chromatin, histone dynamics play a key role in modulating fork progression [7,8]. It is known that DNA replication requires simultaneous separation and reassembly of chromatin, and prolonged disruption and/or inhibition of replication carries a high risk of altering the newly formed chromatin in a way that may alter epigenetic information; these, in turn, may affect gene regulation and the spatial organization of DNA [9].

Heavy metals (HMs) are natural components present throughout the earth’s crust and are characterized by a relatively high atomic mass and high density [10]. HMs can be divided into two groups: (i) elements necessary for the proper growth and development of plants, such as cobalt (Co), copper (Cu), zinc (Zn), magnesium (Mg), molybdenum (Mo), nickel (Ni), copper (Cu), and selenium (Se), which regulate enzymatic functions and (ii) non-essential elements of recognized toxicity, such as lead (Pb), cadmium (Cd), or chromium (Cr) [3,11,12]. While the first group of elements (in small amounts) is important and necessary for the life cycle of plants, high concentrations of both types of HMs are very dangerous. They induce cytotoxic, genotoxic, and mutagenic effects, which may lead to inhibition of plant growth, reduction in nutrient uptake, and, ultimately, to chlorosis and necrosis [3].

HM poisoning in plants brings about an increased production of reactive oxygen species (ROS), which results in oxidative stress (OS) and a consequent lipid peroxidation, membrane breakdown, biological destruction of macromolecules, cell wall injury, ion leakage, and DNA damage [13,14,15]. One of the defence strategies used by plants against HMs is aimed at the strengthening of the antioxidant systems that can counteract OS [13,16]. Another defence mechanism adopted by plants relies on the hindered penetration of HMs by binding them to organic acids, thus trapping them in an apoplastic environment [17], or by anionic groups present in the cell walls of the roots [18,19]. In consequence, most of the HMs entering plants are detoxified as a result of complexation with amino acids, organic acids, or metal-binding peptides [20]. This, in turn, significantly hinders the translocation of HMs absorbed by plant ground organs [11].

DNA repair and replication involve the incorporation of newly synthesized histones to restore the functional chromatin environment [21]. An important aspect during these processes is the epigenetic modification of newly synthesized histones before their incorporation into chromatin. Posttranslational modifications of histone H3 are one of the most extensive epigenetic changes that occur among the four core histones. Generally, attention has been paid to the phosphorylation at a conserved serine residue (Ser10) in the N-terminal tail of histone H3 (H3S10), which is linked to both activation of transcription [22] and to mitotic and meiotic chromosome condensation [23,24,25]. Furthermore, histone H3 acetylations, primarily at lysine 9 (H3K9), 14 (H3K14), and 27 (H3K27), were observed to be enriched near the transcription start sites and in transcribed promoters associated with active transcription [26,27]. The newly synthesized molecules of H3 histones undergo acetylation prior to deposition on DNA, and the preferential acetylation sites differ between species (e.g., H3K56Ac in budding yeast or H3K14Ac and H3K18Ac in human cells [28]). Acetylation helps to attenuate histone H1 deposition and thus prevents chromatin thickening during the replication process [29]. The acetylation of histone H3 on lysine 56 (H3K56Ac) is recognized as a marker of newly synthesized H3 molecules [21,30].

In addition to H3K56 acetylation, one of the best-known fingerprints of the double strand breaks (DSBs) is phosphorylation of histone H2AX on Ser139 (γ-H2AX). RS triggers a set of cellular signaling mechanisms to single and double stranded DNA damage sites at stalled or slowed down replication forks. Using γ-H2AX immunofluorescence labeling, our previous experiments [31] showed an increased number of visible γ-H2AX foci in *V. faba* cells exposed to 24 h incubation with CdCl_2_ and a decreased level of H2AX phosphorylation in the control root cells (after post-cadmium recovery in water). While γ-H2AX is primarily a DSBs marker, ample evidence points to a more general role of γ-H2AX during RS, both in inhibiting replication forks in the absence of DSBs and in repairing collapsed replication forks [9]. Moreover, inhibition or delay of cell cycle progression may extend the time necessary for the initiation of gene expression related to the synthesis and activation of DNA damage repair factors [32,33].

While some studies have shown that H3K56 acetylation is involved in transcription [34,35], other data proved that it is also involved in the response to DNA damage during replication [21,36,37,38]. Confirmation of this important function during DNA biosynthesis is supported by the fact that H3K56Ac levels increase during the S phase and largely decline during the G2/M phase [21].

Another type of epigenetic modification involved in the control of transcriptional elongation, DNA replication, DNA repair, and heterochromatin maintenance is represented by differential mono-, di-, and tri-methylation of H3 histone at lysine K79 (H3K79Me, H3K79Me2, and H3K79Me3). H3K79Me2 was shown to exhibit the highest genome-wide enrichment for replication initiation events observed for any chromatin modification [39]. Accordingly, it seems probable that the prevention of H3K79 dimethylation affects regulatory processes which are responsible for modulating the order and timing of DNA replication. Moreover, H3K79Me2 associates with some replication origins and marks replicated chromatin during the S-phase to prevent re-replication and preserve genomic stability.

Our present studies on *Vicia faba* root meristem cells aimed to analyse and to update our understanding of issues related to changes in epigenetic patterns after exposure to one of the heavy metals—cadmium (CdCl_2_). DNA methylation patterns and H3 histone modifications examined by means of immunofluorescence labeling, related mainly to the process of DNA replication, included: (i) acetylation of histone H3 on lysine 56 (H3K56Ac), a process involved in transcription [34,35], S phase transition [30], and in the response to DNA damage during DNA biosynthesis [21,36,37,40,41]; (ii) the dimethylation of histone H3 on lysine 79 (H3K79Me2), which is correlated with the replication initiation sites [39]; (iii) the phosphorylation of histone H3 on threonine 45 (H3T45Ph), which is involved in DNA replication and apoptosis [42,43].

## 2. Materials and Methods

### 2.1. Plant Material

Sterile seeds of field bean (*Vicia faba* L. subsp. *minor*) were sown in trays, lined with moist blotting paper, and germinated in the dark at 20 °C. After 72 h, plants with roots approximately 2.5 cm long were placed in distilled water (control) and 150 µM CdCl_2_ solution. The concentration of CdCl_2_ was selected based on the available literature data, e.g., [44,45], based on the results obtained in a series of preliminary tests and applied in earlier studies [31]. Cultures were aerated by gentle rotation of Petri dishes in a water-bath shaker (60 r.p.m.), performed for 24 h in the dark.

### 2.2. Feulgen Staining and DNA Cytophotometry

Apical root fragments were fixed for 60 min in cold Carnoy’s mixture (absolute ethanol and glacial acetic acid; 3:1, *v*/*v*). After washing with ethanol (3 times), root tips were rehydrated, hydrolysed for 1 h in 4M HCl, and Feulgen-DNA-stained using the standard method [46] with pararosaniline (applied to selectively stain DNA in a quantitative colorimetric measurement of aldehydes in the Schiff’s test; Sigma-Aldrich, Poznan, Poland). Following 3 times rinsing with SO_2_ water (sulphurous acid prepared from sodium metabisulphite and dilute HCl to stop Feulgen staining, to fix the color, and to elute unbound molecules) and after washing with distilled water, root tips were crushed on microscope slides in 45% acetic acid. Squash preparations were frozen on dry ice, dried, and embedded in Canadian balsam. The total number of cells analysed was always 8000 (out of 10 root meristems) at each time point. Selected cells were photographed with E-600 epifluorescence microscope (Nikon). The extinction of Feulgen-stained cell nuclei was measured at 565 nm with a Jenamed 2 microscope (Carl Zeiss, Jena, Germany) and calibrated in arbitrary units (a.u.). A computer system (Forel, Łódź, Poland) was used for image analysis. Approximately 5000 nuclei were collected to evaluate the distribution of the DNA content.

### 2.3. Immunocytochemical Detections of Dimethylation of Histone H3 on Lysine 79 (H3K79Me2), Acetylation of Histone H3 on Lysine 56 (H3K56Ac), and Phosphorylation of Histone H3 on Threonine 45 (H3T45Ph)

Root meristems from *V. faba* cuttings were fixed (20 min, 20 °C) in PBS (Phosphate-Buffered Saline)-buffered 3.7% paraformaldehyde; then, cell nuclei were isolated, dropped onto glass slides, and air dried (procedure according to [31]). After rinsing thoroughly with PBS, apical parts of roots were crushed on Super Frost Plus slides, dried, and pre-treated with PBS-buffered 8% BSA and 4% Triton X-100 (Sigma-Aldrich, Poznan, Poland) for 50 min (20 °C). Then, slides were incubated with:(1)rabbit polyclonal anti-histone H3K79Me2 antibodies (Abcam, dilution of 1:200);(2)rabbit monoclonal anti-histone H3K56Ac antibodies (Abcam, dilution of 1:200);(3)rabbit polyclonal anti-histone H3T45Ph antibodies (Abcam, dilution of 1:100).

All antibodies were dissolved in PBS containing 1% BSA. Incubations were carried out in a humid atmosphere at 4 °C for 16 h. After washing with PBS, slides were incubated for 1 h (20 °C) with secondary goat anti-rabbit conjugated to Alexa Fluor^®^488 antibodies (1:500; Cell Signaling, Warsaw, Poland) and counterstained with PI (propidium iodide; 0.3 mg mL^−1^). Slides washed with PBS were air dried and embedded in PBS mixture/glycerol (9:1) with 2.3% DABCO.

### 2.4. Immunocytochemical Staining for DNA Methylation

Cut root meristems of the control and CdCl_2_-treated seedlings were fixed for 20 min at 4°C in freshly prepared 4% paraformaldehyde. After fixation, root tips were washed with distilled water, then cell nuclei were isolated, dropped onto glass slides, and air dried. Slides were treated with 4 M HCl for 1.5 h to partially denature the nuclear DNA (room temperature), washed with Tris buffer (pH 7.4) containing 0.5% Triton X-100, and incubated with mouse monoclonal anti-5-methylcytosine antibodies (5-MetC, Sigma-Aldrich, Poznan, Poland) diluted in Tris buffer (1:100). After 16 h (4 °C), slides were washed (Tris buffer) and incubated for 1 h with secondary goat anti-rabbit antibodies conjugated to the bright, green-fluorescent dye Alexa Fluor^®^488 (1:500; Cell Signaling, Warsaw, Poland), washed in Tris buffer, and embedded in PBS mixture/glycerol with DABCO.

### 2.5. Observations and Analyses

Fluorescence intensity analyses, the line plots, and Interactive 3D Surface Plots were performed with the ImageJ software (v1.8.0). Observations were made using E-600 epifluorescence microscope (Nikon) equipped with phase-contrast optics, U2 filter (UVB light; λ = 340–380 nm) for DAPI, B2 filter (blue light; λ = 465–496 nm) for Alexa Fluor^®^488, and G2 filter (green light; λ = 540/25 nm) for PI-stained cell nuclei. All quantitative analyses and nuclear DNA fluorescence measurements were made after converting colour images into greyscale images and were expressed in arbitrary units as mean pixel value (pv) spanning the range from 0 (dark) to 255 (white). The obtained data were expressed as the mean values ± standard deviation (SD). Student’s *t*-tests for paired data were used to compare individual variables. Results used to calculate mean values and micrographs were obtained from one out of three independent experiments (all of them have provided similar sets of data).

## 3. Results

### 3.1. Effect of CdCl_2_ on Chromatin Condensation and Root Meristem Cell Cycle Populations

Microscopic images of the interphase cell nuclei from the CdCl_2_-treated root meristems revealed evident changes in the chromatin structure. In comparison with the control plants, DNA-Feulgen staining showed strong condensation of chromatin, blurring the natural diversity of its fibrillar structures and a distinct, spotted pattern of heterochromatin (Figure 1A).

Cytophotometric measurements of nuclear DNA content revealed quite similar quantities of 2C (G1-phase), 2-4C (S-phase), and 4C (G2-phase) cells both in the control (Figure 1B; dark blue histogram) and CdCl_2_-treated plants (Figure 1C; dark red histogram). After the application of CdCl_2_, a slight shift to the right of the maximum values for the G1- and G2-phase cell cycle subpopulations was observed. This result is probably due to the increased level of chromatin condensation. It can be noted (Figure 1D) that an increase in the number of the G1 phase cells in CdCl_2_-treated roots (30.1% compared to 26.1% in the control) is accompanied by a slight increase in the number of cells in the S (44.8% compared to 41.7%) and a more significant rise in the number of G2 phase cells (25.1% compared to 32.2%; Figure 1B–D).

### 3.2. Dimethylation of Histone H3 on Lysine 79 (H3K79Me2)

Our previous experiments on *V. faba* root meristems treated with 150 µM CdCl_2_ showed that H4 acetylation of lysine 5 in nucleosomal histone H4 (H4K5Ac) is mainly related to DNA replication [31]. It was observed that the variable relationships between H4K5Ac acetylation and successive stages of the cell cycle were strongly modified by cadmium-induced stress conditions. In this work, microfluorimetric measurements were carried out towards evaluating the extent of histone H3 lysine 79 dimethylation (H3K79Me2) (Figure 2). The analyses included: (i) fluorescence intensity (FI) of the entire areas of cell nuclei (together with the nucleoli) during the G1, S, and G2 phases of the cell cycle; (ii) FI of the nuclear chromatin (excluding the nuclear area) and of the nucleoli. Numerous small foci were observed in cell nuclei derived from the control meristems (Figure 2A–B’). The nucleoli were often surrounded by a circle of dots (Figure 2A). In the case of cadmium-treated plants, the foci of fluorescence were also revealed, although they were less numerous than in the control (Figure 2C,D). Another difference that can be observed in the case of *V. faba* seedlings incubated with CdCl_2_ is the much stronger fluorescence of the nucleolar regions (visible against the background of much less fluorescent areas of the extranucleolar chromatin; Figure 2C–D’). As compared with the control, a significant decrease in the level of nuclear fluorescence was noted in all phases of the CdCl_2_-treated interphase cell nuclei (Figure 2E), both in the area of the nuclear chromatin and nucleolus (Figure 2F). The differences in the structure of the nucleus and the nucleolus can be seen in the line plots shown in Figure 2A’–D’.

### 3.3. Acetylation of Histone H3 on Lysine 56 (H3K56Ac)

Using specific antibodies, we analysed another modification—acetylation of histone H3 on lysine 56 (H3K56Ac). We found that, both in the control and CdCl_2_-treated roots, H3K56Ac is distributed in the nucleus and the nucleolus in a punctate (focal) pattern (Figure 3). The immunofluorescence analysis included: (i) the mean total number of intranuclear H3K56Ac foci during the G1, S, and G2 phases and (ii) the mean number of H3K56Ac foci in the nuclear chromatin (excluding the nucleolar area) and in the nucleolus. In control roots, cell nuclei having 2, 2-4C, and 4C DNA content are characterized by the presence of fluorescing H3K56Ac foci situated in the area of the nuclear chromatin (Figure 3A–H) and, especially after treatment with CdCl_2_, at the border of the nucleolus and perinucleolar chromatin (Figure 3F,G). Moreover, quantitative analyses, including Interactive 3D Surface Plots, revealed a greater number of so-called peaks (corresponding to H3K56Ac foci) in the nuclei of meristematic cells incubated in CdCl_2_ (Figure 3E’–H’).

The data calculated for G1-, S-, and G2-cells indicate that 24 h incubation with CdCl_2_ results in an increase in the mean number of intranuclear foci of H3K56Ac histones; the highest, almost a 2-fold increase, was observed in the S-phase (Figure 3I). Results, averaged for all phases, show that the mean number of H3K56Ac foci in the nucleus increased from 16% in the control to 20% in CdCl_2_-treated cells (Figure 3J) and, at the same time, a 2.5-fold increase in the mean number of H3K56Ac foci localized in the nucleolar area (Figure 3J).

### 3.4. Phosphorylation of Histone H3 on Threoninie 45 (H3T45Ph)

Control and cadmium-treated root meristem cells of *V. faba* were immunolabeled with anti-H3T45Ph antibody (Figure 4 and Figure 5A–F) and the following parameters were evaluated: (i) fluorescence intensity (FI) of the entire areas of cell nuclei (together with the nucleoli) during G1, S, and G2 (Figure 5G); (ii) FI of the nuclear chromatin (excluding the nucleolar area) and the nucleolus (Figure 5H).

Root meristem cells of *V. faba* stained with H3T45Ph antibody revealed a plethora of nuclear and nucleolar labeling patterns (Figure 4). In most cases, strong fluorescence of the nucleoli was combined with more or less intense staining of nuclear chromatin (Figure 4A–E’ in control and Figure 4G,G’ in CdCl_2_-treated seedlings); the nucleoli were often surrounded by a circle of fluorescing dots (Figure 4H–I’). Moreover, antigenic determinants of H3T45Ph proteins were found, localized within the internal areas of interphase chromatin (Figure 4C–K’).

The line plots passing the nucleus throughout the nucleolar region revealed a significantly higher degree of fluorescence, mostly in the cadmium-treated seedlings (Figure 5A’–F’), than that emitted by the extranucleolar chromatin region (Figure 5A–F). As compared to the control, the 24 h incubation with CdCl_2_ resulted in a 20% decrease in FI in the G1, S, and G2 phases of the cell cycle (Figure 5G). The second analysis revealed a decrease of about 15% in the FI values calculated both for the nuclear and nucleolar chromatin (Figure 5H).

### 3.5. Immunocytochemical Detection of 5-Methylcytosine (5-MetC)

The distribution and levels of methylated DNA (5-methylcytosine; 5-MetC) were observed in the entire areas of cell nuclei (together with the nucleoli) in both the control and CdCl_2_-treated root meristems of *V. faba*.

Compared to the control (Figure 6A–D), the intensity of 5-MetC immunofluorescence in roots exposed to cadmium (Figure 6E–G) decreased by about 30% (Figure 6H). It was noted that the regions of significantly increased fluorescence throughout the interphase correspond with the margins of the nucleoli, quite often visible as a ring of smaller or larger dots (Figure 6A–F). In the mitotic stages, especially in the root meristem cells treated with CdCl_2_ (Figure 7), the chromosomes show particularly strong fluorescence signal in the form of bands (in different parts of the chromosome arms), localized in the pericentromeric and intercalary regions (Figure 7A–D).

## 4. Discussion

Numerous studies have demonstrated a significant role of epigenetic mechanisms in the regulation of plant response to HM-induced stress conditions [3,47,48]. These mechanisms include three interconnected processes that work together in gene regulation: DNA methylation, histone modification (such as acetylation, methylation, phosphorylation, ubiquitination, biotinylation, and sumoylation), and regulatory functions mediated by non-coding RNAs [49,50]. It has been shown that: (i) epigenetic marks are used to change signaling pathways which alter cellular metabolism and, eventually, allow plants to protect themselves from possible DNA damage caused by HMs; (ii) high levels of methylation can protect DNA from endonuclease cleavage, which increases the resistance of cells to HMs; (iii) epigenetic changes serve to regulate genes responding to stress factors [3].

Our current experiments on *V. faba* root meristem cells exposed to 150 µM cadmium chloride (CdCl_2_) concentrated mainly on histone H3 modifications and epigenetic patterns of DNA methylation that are both known to be associated with the DNA replication stress conditions (RS). Accordingly, as an appropriate tool to resolve the first issue, immunofluorescence studies were chosen for the purpose of evaluating: (i) histone H3 acetylation on lysine 56 (H3K56Ac), which is also a key epigenetic histone modification appearing in response to DNA damages; (ii) histone H3 dimethylation on lysine 79 (H3K79Me2); (iii) histone H3 phosphorylation on threonine 45 (H3T45Ph).

In the control seedlings of *V. faba*, the mean number of intranuclear foci of H3K56Ac histones was low in all phases of the cell cycle. In contrast, cells exposed to 150 µM CdCl_2_ revealed a significant increase in the mean number of fluorescing foci, which, as demonstrated earlier after exposure to cadmium [31], seems to be associated with the appearance of DNA damage. Therefore, the obtained results confirm the theory of Masumoto et al. [21] that cells with DNA breaks exhibit a high level of H3K56 acetylation, and that the maintenance of this modification depends on DNA damage checkpoint proteins that promote the DNA repair mechanisms. Therefore, acetylation of histone H3K56 creates a favourable chromatin environment for DNA repair, and a key element in response to the DNA damage.

A distinct fraction of the genomic replication initiation sites is associated with a high frequency of dimethylation of histone H3 on lysine 79 (H3K79Me2) [39]. H3K79 undergoes mono-, di-, and tri-methylation by the enzyme DOT1 (Disruptor of Telomere Silencing 1, [50]). DOT1-like, or DOT1L in humans [51], is involved in transcriptional elongation, DNA repair, and heterochromatin maintenance. H3K79Me2 is particularly abundant in the late S phase [52], and proper functioning of DOT1L, in cooperation with H2B ubiquitination, promotes the DNA damage checkpoint [39,53]. To investigate the possible association of H3K79Me2 with initiation of DNA replication, we measured H3K79Me2 enrichment levels during the G1, S, and G2 phases of the cell cycle in the control and CdCl_2_-treated cells. In untreated seedlings, the mean value of H3K79Me2 fluorescence was highest during DNA replication and maintained (albeit at a lower level) in the G1 and G2 phases of the cell cycle. The presence of H3K79Me2 in all phases of the cycle, not only in the S phase, can be explained by the theory of Fu et al. [39] that, during the G1 phase, the H3K79Me2 modification occurs in regions proximal to the replication origin and then extends to adjacent regions of chromatin during DNA synthesis. Then, H3K79Me2 marks, again, clusters corresponding to the initiation sites in G2, suggesting that S phase-specific marks that are not related to DNA replication origins may be removed and that the origin-specific marks remain after mitosis for the next cell cycle [39]. We also observed a significant decrease in the level of H3K79Me2 fluorescence in all cells after CdCl_2_ treatment, which is probably related to the decrease in the rate of DNA replication and/or to the appearance of reactive oxygen species (ROS) [31]. This result corresponds well with other studies showing that ROS can interfere with S phase progression by slowing down the rate and/or inhibiting the movement of replication forks [54].

Another important modification that occurs concurrently with DNA replication is the phosphorylation of histone H3 on threonine 45 (H3T45Ph), which regulates numerous nuclear functions including transcription, DNA damage repair, mitosis, and apoptosis in the budding yeast *Saccharomyces cerevisiae* [42]. In the T45A mutant, the prolonged RS was induced by MMS (methyl methanesulphonate) by accumulation of H3T45 phosphorylation over time. As shown in our studies, 24 h incubation with CdCl_2_ induced changes in the cell cycle-related H3T45Ph. Our previous research [31] revealed that acetylation of lysine 5 in histone H4 (H4K5Ac) is related to the transcriptional activity and inversely correlated with nucleolar rDNA replication in the early stages of the S phase. The pattern of histone H4 acetylation, especially as regards the nucleolar region, shows a striking similarity to the distribution of H3T45Ph fluorescence. The spread of epigenetic markers from the G1 phase to the G2 phase in the control root meristem cells of *V. faba* allows us to suggest that phosphorylation of H3T45 may play an important regulatory role in both DNA replication and transcription. The 24 h incubation with CdCl_2_ resulted in a slight decrease in fluorescence in the G1, S, and G2 phases of the cell cycle. Therefore, it cannot be ruled out that cadmium may erode the epigenetic marks observed throughout interphase [31]. Moreover, our analyses revealed different labeling patterns of nucleolar chromatin, both in the control and after incubation of the seedlings in cadmium. A slight decrease in fluorescence within the nucleolar region may indicate its lesser susceptibility to cadmium-induced stress [31]. It should be noted, furthermore, that, according to Jasencakova et al. [55], nucleolar territories are devoid of nucleosomes.

The chromatin landscape becomes dynamic in response to environmental signals, thus modulating DNA accessibility to factors that regulate gene expression [56]. DNA methylation is the best documented epigenetic modification involved in a number of molecular processes, such as activity of transposable elements, alternative splicing, regulation of temporal, and spatial gene expression [3,57]. Moreover, it is involved in various biological processes such as flowering and flower and leaf morphogenesis, as well as fertility regulation through gene silencing [58,59]. The phenomenon of DNA hypo- or hypermethylation, appearing in response to various stresses, regulates gene expression and the subsequent physiology of plants [59,60]. The level of DNA methylation changes in response to various metals in white clover (*Trifolium repens* L.), industrial hemp (*Cannabis sativa* L.), rape (*Brassica napus*), radish (*Raphanus sativus* L.), and rice (*Oryza sativa*) [61,62,63,64].

One of the aims of our present study was to analyse and compare the level of DNA methylation in the control (untreated) and cadmium-treated nuclei. An immunostaining method using specific antibodies against 5-MetC-modified DNA was used to determine the presence and level of DNA methylation. Compared to the control, the level of immunofluorescence of DNA methylation in roots exposed to cadmium decreased by about 30%. The literature data report that Cd (cadmium), Ni (nickel), and Cr (chromium) cause oxidative damage [65,66,67], which induces DNA hypomethylation. Our earlier work [31] showed that 24 h treatment of seedlings with CdCl_2_ significantly increased H_2_O_2_ production. Similar results were obtained previously in studies on roots in alfalfa [68], soybean [69], BY-2 tobacco cells [70], and other plant models. Therefore, it cannot be ruled out that the phenomenon of DNA hypomethylation is related to DNA damage induced by ROS. The mechanisms by which ROS generates hypomethylation are different: (1) ROS activates endonuclease, with the consequent induction of single-stranded breaks that make DNA a weak acceptor of methyl groups; (2) metal-mediated ROS production in the vicinity of DNA generates the premutagenic adduct oxo8dG (8-oxo-20-deoxyguanosine [61,71]), resulting in a strong inhibition of methylation of adjacent cytosine residues; (3) oxidative stress induces an increase in nicotinamide (NIC) levels and poly (ADP-ribose) polymerase-1 activity [72]. In plant tissues, NIC is metabolized to trigonellin (N-methylnicotinic acid, TRIG), which has a hypomethylating effect and can induce a protective metabolism at the gene level [73]. Thus, hypomethylation can either be a precise defence mechanism by which the cell regulates gene expression, or an indirect effect of metal stress.

A number of studies have shown that DNA modifications and post-translational histone modifications play an important role in gene expression and plant development under stress factors. Two epigenetic strategies are important in plant response to heavy metal-induced stresses, including effects exerted by cadmium: (i) epigenetic markers serve as a mechanism to protect plants from possible metal ion-induced DNA damage due to random DNA methylation along the genome, and (ii) epigenetic changes are used to regulate the expression of stress-responsive genes. Some of these stress-induced modifications reset to baseline when the stress factor is gone, while some of them can be transferred as “stress memory” and inherited by the cell [74]. Thus, epigenetic memory of stress may enable plants to more effectively deal with subsequent stress factors.

## Figures and Tables

**Figure 1 cells-10-03409-f001:**
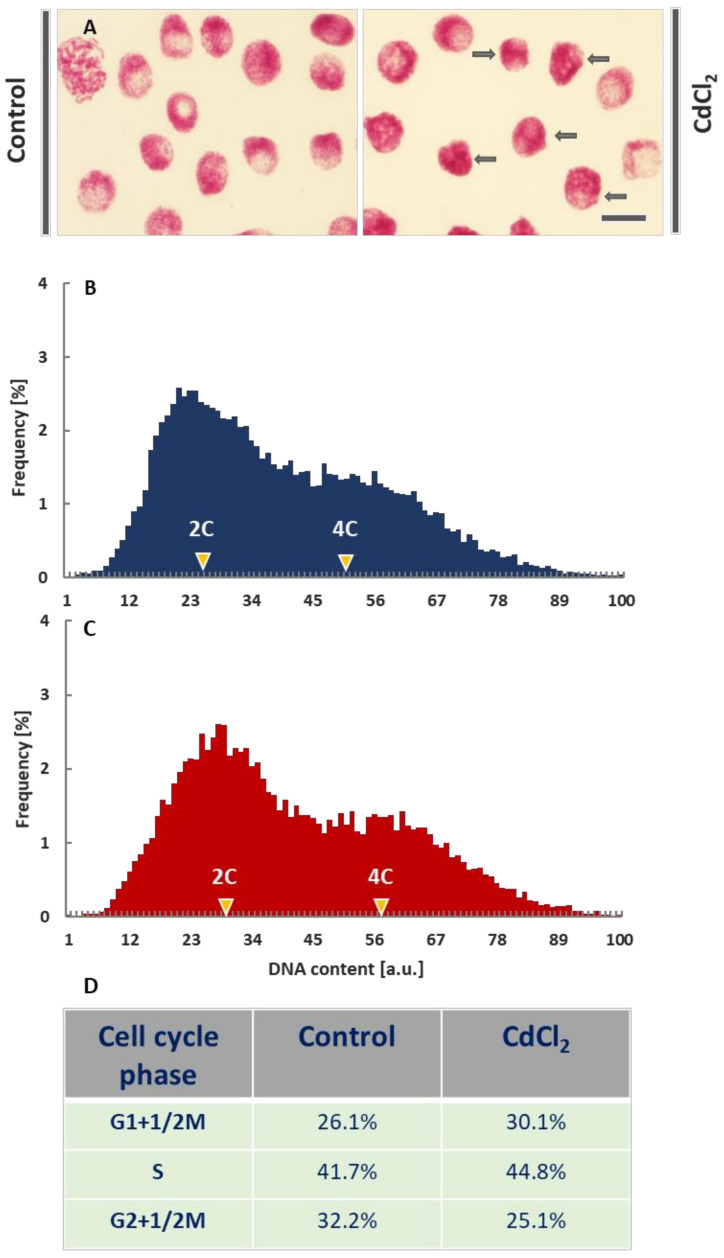
Feulgen-DNA staining of *V. faba* cell nuclei from the control root meristems and after incubation with CdCl_2_ (scale bar = 20 μm); arrows in the micrographs (**A**) show cell nuclei with significantly condensed chromatin. Histograms of frequency distribution of nuclear DNA content (arbitrary units, au) in control meristems (**B**) and after 24-h incubation of root meristems with CdCl_2_ (**C**). Each population consisted of more than 18,000 cell nuclei. The table below the histograms (**D**) shows estimated frequencies (%) of G1+½M phase cell nuclei (including of ana- and telophases), S phase cell nuclei, and G2+1/2M phase cell nuclei (including pro- and metaphases) in the control and CdCl_2_-treated root meristems.

**Figure 2 cells-10-03409-f002:**
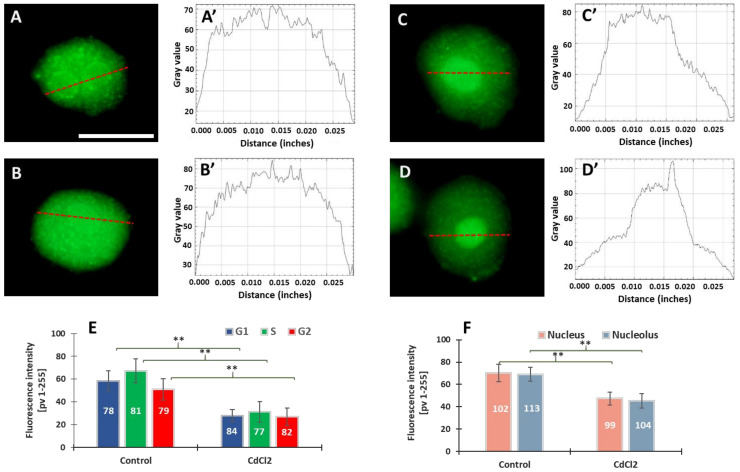
Immunofluorescence detection of H3K79Me2 in the *V. faba* cell nuclei (represented by the G2 phase), from the control (**A**,**B**) and CdCl_2_-treated seedlings ((**C**,**D**); scale bar = 10 μm). Densitometric plots showing changes in the intensity of fluorescence of the nuclear and nucleolar regions in the control (**A****’**,**B’**) and after CdCl_2_ treatment (**C’**,**D’**); the course of each plot is marked by a red dashed line. Mean FI of H3K79Me2 in the interphase cell nuclei (**E**) and mean FI of H3K79Me2 in the nuclear chromatin and nucleolar regions (**F**) in the control and CdCl_2_-treated root meristem cells. Error bars represent standard deviation (SD). The numbers of cell nuclei/nucleoli used for immunostaining quantification are given inside the diagram bars. Statistical significance between mean values (at ** *p* < 0.01, indicated by a black asterisks) was assessed with Student’s *t*-test.

**Figure 3 cells-10-03409-f003:**
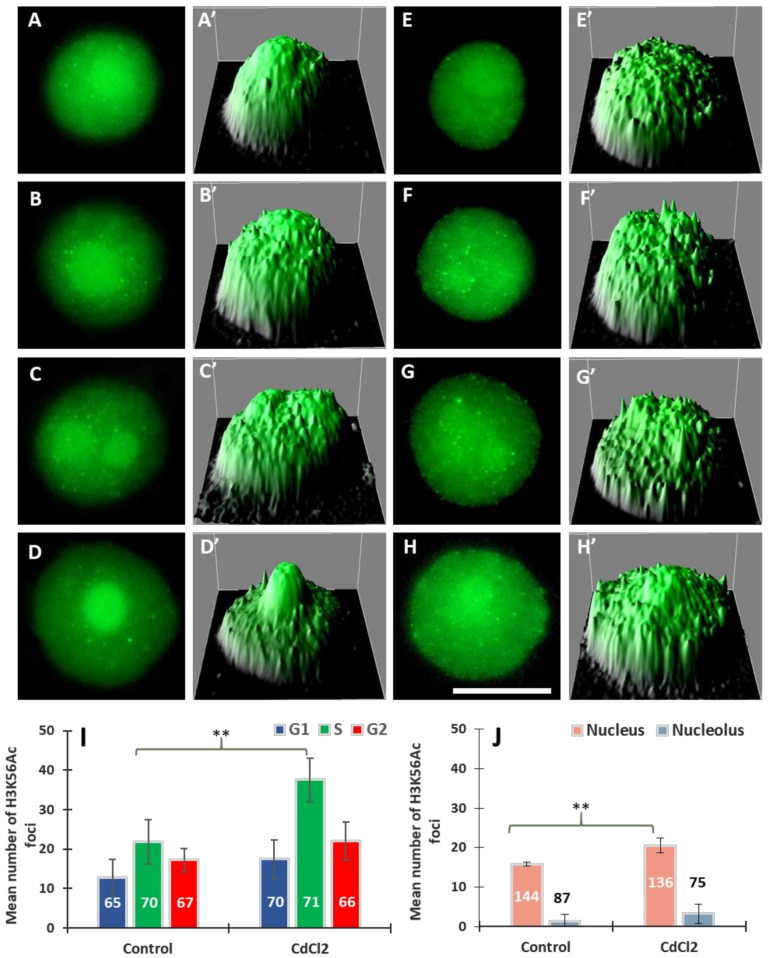
Immunofluorescence of H3K56Ac in cell nuclei from the control (**A**–**D**) and CdCl_2_-treated root meristems (**E**–**H**) in the G1 (**A**,**E**), S (**B**,**C**,**F**,**G**), and G2 phases (**D**,**H**); scale bar = 10 μm. At the right side of each micrograph are corresponding interactive 3D surface plots (**A’**–**H’**). Mean number of H3K56Ac foci generated in G1, S, and G2 phase nuclei in the control and CdCl_2_-treated root meristem cells of *V. faba*, (**I**) and mean number of H3K56Ac foci in the nuclear and nucleolar chromatin from the control and CdCl_2_-treated root meristems (**J**). Error bars represent standard deviation (SD). The numbers of cell nuclei used for immunostaining quantification are given inside or above the bars. Statistical significance between mean values (at ** *p* < 0.01, indicated by a black asterisks) was assessed with Student’s *t*–test.

**Figure 4 cells-10-03409-f004:**
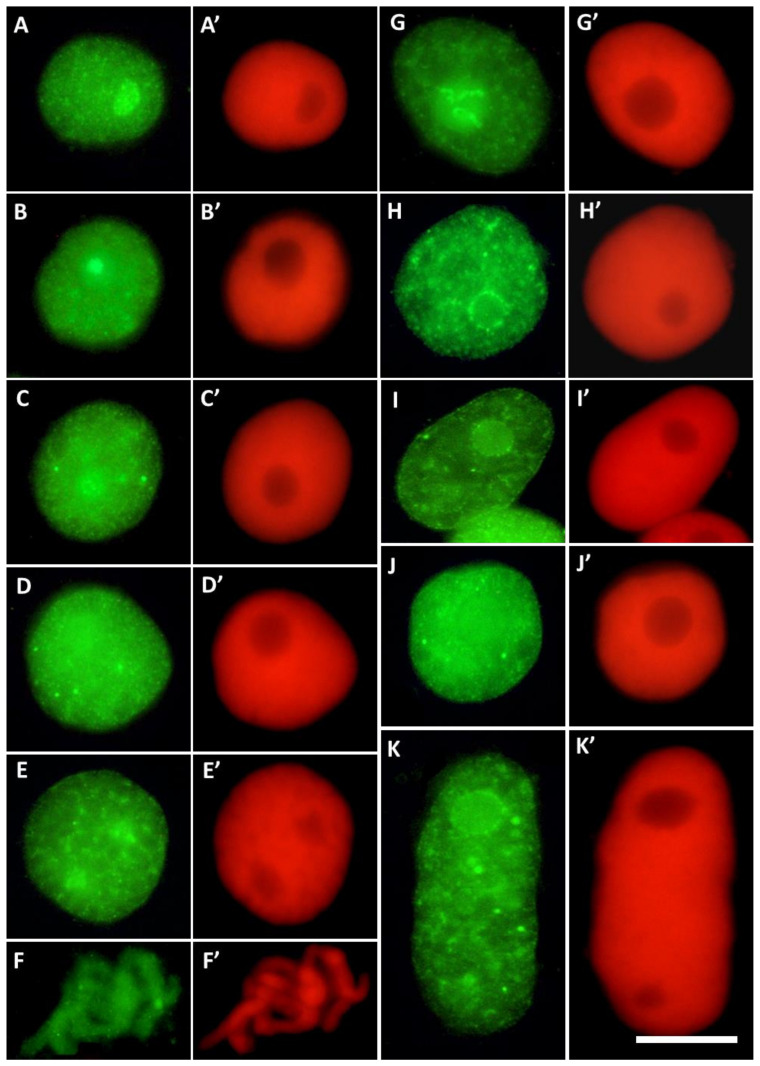
Different labeling patterns observed in cell nuclei after immunostaining with antibodies against H3T45Ph conjugated with Alexa Fluor^®^488 ((**A**–**K**); green fluorescence) and corresponding images of cell nuclear DNA stained with propidium iodide ((**A’**–**K’**); red fluorescence). The control cells (**A**–**F’**) revealed predominant nucleolar staining of H3T45Ph (**A**–**B’**), localization of H3T45Ph within the internal areas of interphase chromatin (**C**–**D’**) and prophase (**E**,**E**’), and labeling of the chromosomal areas corresponding to the nucleolus-organizing regions (NORs) during prometaphase (**F**,**F’**). The CdCl_2_-treated cells show predominant nucleolar staining in cell nuclei (**G**,**G’**), specific labeling around nucleoli (**H–I’**), and localization of H3T45Ph within the internal areas of interphase chromatin (**J**–**K’**).

**Figure 5 cells-10-03409-f005:**
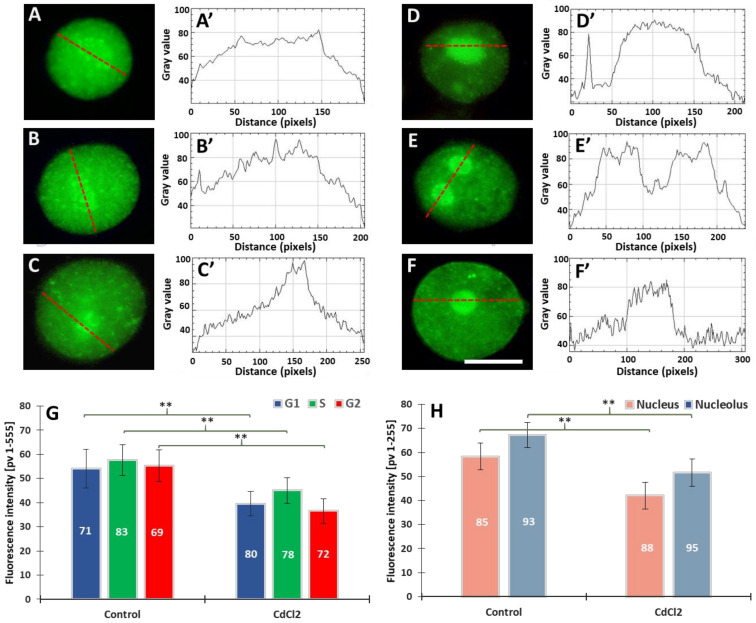
Immunofluorescence detection of H3T45Ph in cell nuclei from the control (**A**–**C**) and CdCl_2_-treated seedlings of *V. faba* (**D**–**F**), in the G1 (**A**,**D**), S (**B**,**E**), and G2 phases (**C**,**F**); scale bar = 10 μm. Densitometric plots showing changes in fluorescence in the nuclear and nucleolar regions in the control (**A****’**–**C’**) and after treatment with CdCl_2_ (**D’**–**F’**); the course of each plot is marked by a red dashed line. Mean H3T45Ph fluorescence intensity of G1, S, and G2 phase nuclei in the control and CdCl_2_-treated root meristem cells (**G**) and mean intensity of H3T45Ph fluorescence of the nuclear and nucleolar chromatin in the control and CdCl_2_-treated root meristems (**H**). Error bars represent standard deviation (SD). The numbers of cell nuclei/nucleoli used for immunostaining quantification are given inside the bars. Statistical significance between mean values for the intensities of H3T45Ph fluorescence (at ** *p* < 0.05, indicated by a black asterisks) was assessed with Student’s *t*-test.

**Figure 6 cells-10-03409-f006:**
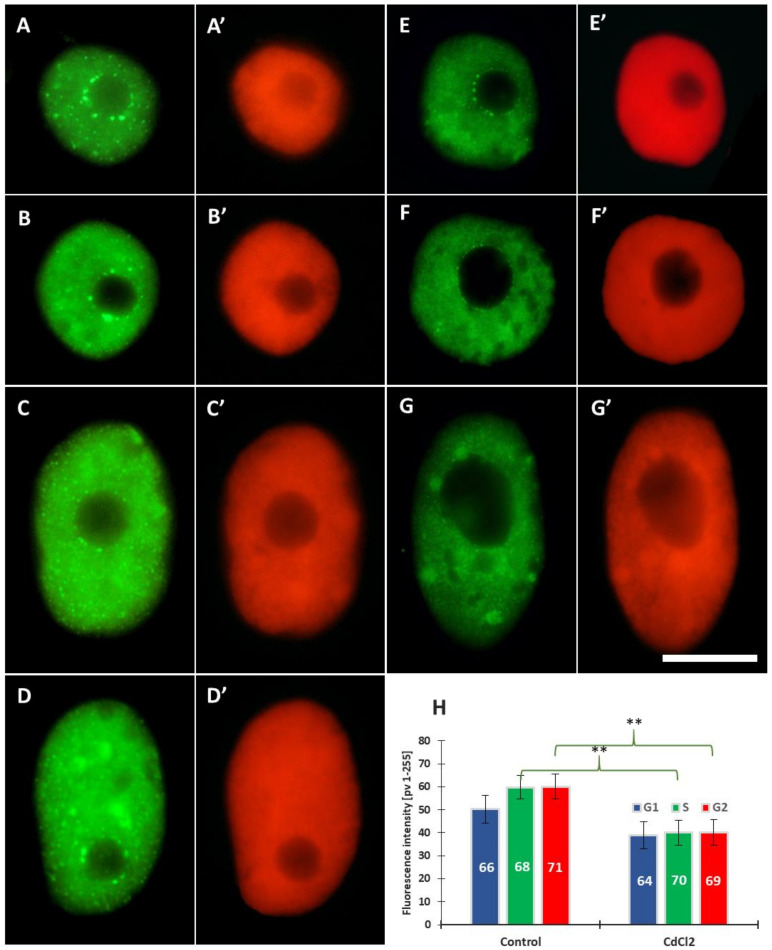
Immunofluorescence detection of 5-MetC in cell nuclei from the control (**A**–**D**) and CdCl_2_-treated root meristems (**E**–**G**) in the G1 (**A**,**E**), S (**B**,**F**), and G2 phases (**C**,**D**,**G**); on the right side of each photo, the same nuclei stained with propidium iodide (**A’**–**G’**); scale bar = 10 μm. Mean FI of 5-MetC in the G1, S, and G2 phase nuclei in the control and CdCl_2_-treated root meristem cells of *V. faba* (**H**). Error bars represent standard deviation (SD). The numbers of cell nuclei used for immunostaining quantification are given inside the bars. Statistical significance between mean values for the intensities of H3T45Ph fluorescence (at ** *p* < 0.05, indicated by a black asterisks) was assessed with Student’s *t*-test.

**Figure 7 cells-10-03409-f007:**
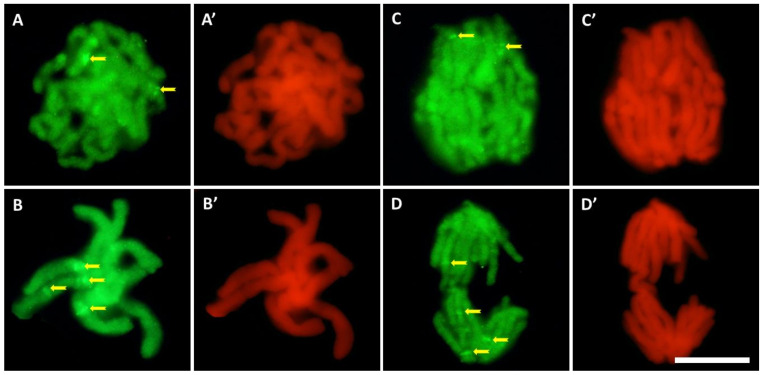
Immunofluorescence detection of 5-MetC during prophase (**A**), metaphase (**B**), and anaphase (**C**,**D**) in CdCl_2_-treated cells; bands in the pericentromeric and intercalary regions (arrows). On the right side of each photo, the same stage of mitosis stained with propidium iodide (**A’**–**D’**); scale bar = 10 μm.

## Data Availability

The data presented in this study are openly available.

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
