# Peer review of "Changes in Epigenetic Patterns Related to DNA Replication in *Vicia faba* Root Meristem Cells under Cadmium-Induced Stress Conditions"

_cells, 2021, doi:10.3390/cells10123409_

Round 1
Reviewer 1 Report
The work addresses Cd toxicity in plants.
There are sticky notes added to the manuscript at places of concern--many of these ask for clarifications of abbreviations etc
I think the manuscript could be reordered for better impact. Much of what I was searching for in the Introduction is found in the initial paragraphs of the Conclusions. Thus when I read the paper I thought the Results were factual but did not understand how what is seen in the studies related to known information especially the consequences of the epistatic changes.
Some of my comments are based on my background with Cd and plants which is not directed towards nuclear changes -- thus I struggled and used google a lot to understand why and what was being shown to me. Some other readers may also need such help.
I am also stuck with the fact that this is clearly a lab study and there is no discussion on what the findings might mean to plants that are suffering from exposure to Cd from polluted conditions. Are you indicating that all of these nuclear changes contribute to the stunted growth?

Author Response
We would like to thank you very much (Reviewer 1) for the comments and remarks which have significantly improved our manuscript. According to your suggestions, the following changes have been made in the text of the successive parts of our manuscript:
ABSTRACT
- Full name Vicia faba in italics was introduced.
- Instead of CdCl2, subscript CdCl2 was applied in all cases.
- The huge sentence was divided into 2 sentences and 3 points were introduced.
- “Stress factors” were changed into “cadmium-induced stress conditions”.
KEYWORDS
- According to our best knowledge, the term “replication stress” or “DNA replication stress” may be used in order to characterize hampered progression of DNA synthesis (e.g., due to slowed down or stalled replication forks). This term was used in our earlier articles.
INTRODUCTION
- According to the Reviewer’s 1 suggestion, selenium (Se) was classified as an element used in some enzymes.
- As suggested, we added that HMs poisoning results in cell wall injury.
- We have added few sentences concerning histone H3 and H3 methylation.
- According to the Reviewer’s 1 suggestions, two fragments of Discussion (first, concerning “replication stress” and second, concerning histone epigenetic modifications) were moved to Introduction.
- Furthermore, we have shifted part of Introduction to Discussion (now, as the first paragraph).
MATERIALS and METHODS
- The term “cutting” was used by mistake and now is changed to “Plants”. Whole roots of faba seedlings (not the excised root tips) were incubated and used in our experiments.
- Information was added that “Roots were aerated by gentle rotation of Petri dishes in a water-bath shaker (60 r.p.m.)”.
- The concentration of CdCl2 was chosen based on the results obtained from available literature data [e.g. Unyayar 2006, Souguir 2011] and a series of preliminary tests. Appropriate information and citations were added to the text.
- Explanations for SO2-water (including preparation of the solution) and for standard Feulgen-DNA staining was added to the text (including original citation). Many protocols are specifically applied for particular plant tissues.
The Feulgen technique selectively stains DNA, and under controlled conditions, can be used for the photometric determination of DNA content. The reaction consists of two steps. Fixed material is treated HCl and afterwards, the material is immediately transferred into Schiff's reagent at room temperature (for at least 30 min or until the tissue stains deep purple) [acc. to . FEULGEN STAINING PROTOCOL, Kansas State University].
- PBS is explained.
- Alexa Fluor®488 is a bright, green-fluorescent dye conjugated to the anytibody molecule. Information in the text was extended.
RESULTS
- Two sentences were changed. In the first, the nuclear DNA contents increasing during interphase progression (expressed as 2C, 2-4C and 4C) were set up with the cell cycle phases (G1, S, and G2); in the second, “cell cycle” was added to the text.
- Figure 1 was improved. Micrographs (part A) and percentages (part D; Table) were corrected and changed.
- Indeed, there are no striking differences between the percentages of G1, S, and G2 cells in the two compared series (control vs. CdCl2). This fact has been noted in the text.
- Legend to Fig. 1 was changed. We hope that the present version of the legends informs about statistical methods and about the significance of the results. Some correction in the text of Materials and Methods has also been done concerning our measurements.
- Color codes in Fig. 3 were corrected.
- Two colors of fluorescence (Fig. 4) is explained (first – green Alexa Fluor®488 fluorochrome conjugated with the antibody, and second – red propidium iodide, a commonly used fluorochrome staining nuclear DNA). This explanation is valid in any presentation with two-colour staining procedure.
- 7. Micrographs showing cell nuclei stained using fluorochrome (Alexa Fluor®488)-conjugated antibodies were labelled using A, B, C, D, etc, and corresponding micrographs showing the same cell nuclei stained with propidium iodide (DNA specific fluorochrome) were labelled using A’, B’ C’, D’, etc.
DISCUSSION
- According to the Reviewer’s 1 suggestions, the paragraph concerning “replication stress” was shifted to Introduction.
- The second paragraph concerning histone epigenetic modifications was moved to Introduction.
- New paragraph (now, the first in the Discussion) was shifted from Introduction.
LITERATURE
- New citations were introduced to our manuscript, in agreement with the suggestions of Reviewer 1 and Reviewer 2.
ADDITIONAL CHANGES
- A new author (Janusz Maszewski), erroneously omitted in the former version, was added.
- Mistakes found in the text were corrected.
All changes introduced to the new version of the manuscript (according to Reviewers’ suggestions) are marked green in a Corrections File and (in some cases) supplied with additional information.
Reviewer 2 Report
In this Manuscript entitled "Changes in epigenetic patterns related to DNA replication in 2 Vicia faba root meristem cells under cadmium-induced stress conditions " Aneta et al investigated effects of Cadmium stress on histone marks and DNA methylation. Authors exposed Vicia faba roots to cadimium and profiled levels of DNA methylation, H3K79Me2, H3K56Ac and H3T45Ph. Authors demonstrated most of these marks are decreased and speculated this could lead to transcriptiuonal variation.
Genereally, this manuscript will be of interest to the audiences studding stress epigenetics in palnts. This manuscript is in good shape for publication in Cells, however couple of minor points should be adressed.
- Authors showd indicated the number of nuclei they used for immnunostaing quantification. This numbber should be indicated on all barcharts (For Control and stress conditions and for Various histone marks.)
- For All barcharts significance is indicated. But authors should metion which test they used to perform statistical analysis in the legends.
- Authors should cite the work of Wibowo and Becker et al 2016 eLife paper ( DOI: 10.7554/eLife.13546 ) in their lines 416 or 418
Author Response
We would like to thank you very much (Reviewer 2) for the comments and remarks which have improved our manuscript. According to your suggestions, the following 3 changes have been made in the text:
- The numbers of cell nuclei (and nucleoli in some cases) in used for immunostaining quantification were added and displayed inside the diagram bars.
- Information about statistical methods and about the significance of the results was added to the legends. Some correction in the text of Materials and Methods has also been done concerning our measurements.
- An article by Wibowo and Becker et al 2016 [eLife paper (DOI: 10.7554/eLife.13546) was cited in the last but one line of Discussion.
ADDITIONAL CHANGES
- A new author (Janusz Maszewski), erroneously omitted in the former version, was added.
- Mistakes found in the text were corrected.
- New citations were introduced to our manuscript, in agreement with the suggestions of Reviewer 2 and Reviewer 1.
All changes introduced to the new version of the manuscript (according to Reviewers’ suggestions) are marked green in a Corrections File and (in some cases) supplied with additional information.
Round 2
Reviewer 1 Report
minor changes see sticky notes
great images

Author Response
Dear Reviewer
Thank you very much for the review of our manuscript and for additional comments. We have modified the text according to your all suggestions (except the last one, because we were unable to grasp the meaning of the commentary). All the changes in the revised version of our manuscript were made using "track changes".
Yours faithfully,
Aneta Żabka and Janusz Maszewski
